# Electrophysiological correlates of symbolic numerical order processing

**Clemens Brunner** *, **Philip Schadenbauer, Nele Schröder, Roland H. Grabner, Stephan E. Vogel**

Department of Psychology, Educational Neuroscience, University of Graz, Graz, Austria

* clemens.brunner@uni-graz.at

## Abstract

Determining if a sequence of numbers is ordered or not is one of the fundamental aspects of numerical processing linked to concurrent and future arithmetic skills. While some studies have explored the neural underpinnings of order processing using functional magnetic resonance imaging, our understanding of electrophysiological correlates is comparatively limited. To address this gap, we used a three-item symbolic numerical order verification task (with Arabic numerals from 1 to 9) to study event-related potentials (ERPs) in 73 adult participants in an exploratory approach. We presented three-item sequences and manipulated their order (ordered vs. unordered) as well as their inter-item numerical distance (one vs. two). Participants had to determine if a presented sequence was ordered or not. They also completed a speeded arithmetic fluency test, which measured their arithmetic skills. Our results revealed a significant mean amplitude difference in the grand average ERP waveform between ordered and unordered sequences in a time window of 500–750 ms at left anterior-frontal, left parietal, and central electrodes. We also identified distance-related amplitude differences for both ordered and unordered sequences. While unordered sequences showed an effect in the time window of 500–750 ms at electrode clusters around anterior-frontal and right-frontal regions, ordered sequences differed in an earlier time window (190–275 ms) in frontal and right parieto-occipital regions. Only the mean amplitude difference between ordered and unordered sequences showed an association with arithmetic fluency at the left anterior-frontal electrode. While the earlier time window for ordered sequences is consistent with a more automated and efficient processing of ordered sequential items, distance-related differences in unordered sequences occur later in time.

## Introduction

People use integer numbers in their daily lives to describe both the number of elements (cardinality) as well as the order of elements (ordinality) of a given set [1]. Numerical order processing pertains to the knowledge that numbers convey information about their relative rank or position within a sequence [2, 3]. Many behavioral and neuroimaging studies have revealed the importance of numerical order processing for concurrent arithmetic skills in children [4–

**Data Availability Statement:** All files, including data and analysis scripts, are publicly available at https://osf.io/7udzb/.

**Funding:** The author(s) received no specific funding for this work.

**Competing interests:** The authors have declared that no competing interests exist.

7] and adults [8–11], as well as its predictive value for future arithmetic skills [12, 13]. Besides the well-known relevance of learning the correct order of number words for the development of arithmetic skills [14], current research suggests that more sophisticated skills like arithmetic rely on a rich semantic network of numerical associations, and that ordinal processing is a key indicator of this development [3]. However, our current understanding of the neural underpinnings related to ordinal processing is limited. While several functional magnetic resonance imaging (fMRI) studies have reported distinct activation patterns in frontal and parietal cortex regions in children [7, 12, 15, 16] and adults [17], electroencephalography (EEG) might reveal complementary insights into ordinal processing due to its high temporal resolution.

To the best of our knowledge, there are only two relevant prior studies on electrophysiological correlates of ordinal processing. In one of these EEG studies, Turconi and colleagues [18] manipulated numerical distance (the difference between two numbers) in two task conditions. In the ordinality condition, participants had to judge if a given number comes before or after 15, whereas in the magnitude condition, they had to decide if a number is smaller or larger than 15. Behavioral results revealed a canonical distance effect, a hallmark of cardinality processing [3], in both conditions. This means that participants responded faster and more accurately when the distance between the two numbers was large rather than small. Likewise, the authors found similar activation patterns when analyzing event-related potentials (ERPs) in both ordinal and magnitude conditions. However, the canonical distance effect in the magnitude condition was strongest over left parietal electrodes within a time window of 170–210 ms after stimulus onset, whereas the effect occurred bilaterally in a slightly later time window of 210–240 ms in the ordinal condition. These findings indicate that magnitude and ordinality processing share common neurophysiological mechanisms, but also seem to recruit specific networks for each task. However, since tasks in both conditions involved only two numbers, it is possible that participants used magnitude comparisons (as suggested by the canonical distance effect) in the ordinality condition and vice versa [19].

Sequences with three numbers prevent this potential issue and lead to specific behavioral characteristics of ordinal processing, such as the reverse distance effect [3]. In this task, participants must decide if three presented numerals are ordered (e.g., 2–3–4) or unordered (e.g., 3–2–4). Response times and error rates show three characteristic behavioral patterns.

First, ordered sequences are recognized faster and more accurately than unordered sequences, which might correspond to efficient verbal [20] and visual-spatial representations of numbers [21]. Some studies have associated individual differences in this pattern with arithmetic skills in children [7, 22] and adults [9].

Second, ordered sequences with an inter-item distance of one (e.g., 2–3–4) are recognized faster and more accurately than ordered sequences with an inter-item distance of two (e.g., 2–4–6), which is known as the reverse distance effect [3, 23, 24]. This effect can be found in children [22] and adults [9, 10, 25]. Although its precise nature is debated [26], some evidence suggests that it may reflect efficient retrieval of familiar sequences from long-term memory [10, 27, 28]. A reverse distance effect also emerges for other ordinal sequences such as the letters of the alphabet or months of the year (e.g., [9, 29]). This finding helps to differentiate it from the canonical distance effect associated with magnitude processing, because non-numeric sequences only convey positional information.

When examining associations with individual differences in arithmetic skills, some studies have found a relationship with the reverse distance effect [8, 10, 25], whereas other studies have failed to provide evidence for such a connection [5, 19, 30]. One possible explanation for this inconsistency might be that individuals process ordered sequences with different strategies, as not all individuals exhibit a reverse distance effect (some even show a canonical distance effect) when processing ordered sequences [25].

Third, unordered sequences with an inter-item distance of two (e.g., 4–6–2) are recognized faster and more accurately than unordered sequences with an inter-item distance of one (e.g., 3–4–2). This canonical distance effect indicates that processing unordered sequences might be related to an iterative mental comparison of number magnitudes [22]. To our knowledge, there are no reports of significant associations between the canonical distance effect in unordered sequences and arithmetic skills.

Overall, much of our existing knowledge about ordinal processing is linked to the three-item order verification task [4, 8–10, 25], suggesting that several different components are involved in this task. This is reflected by the reverse distance effect as a marker for efficient retrieval of positional information in ordered sequences, as well as the canonical distance effect as a correlate of magnitude processing in unordered sequences.

As far as we know, only Rubinsten et al. [31] have employed this three-item task to investigate electrophysiological correlates of ordinality processing in adults. In particular, they used a non-symbolic variation with three dot arrays instead of numerals and manipulated the following properties: ordered vs. unordered, the numerical ratio between neighboring sequences of 0.3 vs. 0.6, and ascending vs. descending order. Behavioral results showed that participants recognized ordered sequences faster than unordered sequences. They also identified descending sequences faster than ascending sequences, potentially because participants were native Hebrew speakers (the language is written from right to left). ERP results revealed significant peak amplitude differences between conditions in three time windows. First, the authors found a larger positive amplitude for ordered sequences compared to unordered sequences at right parietal and lateral occipital electrode sites in a time window of 80–130 ms post stimulus. Such early effects are usually attributed to differences in stimulus parameters such as luminance and spatial frequency, but the authors argue that their finding could also reflect an early process related to automatic non-symbolic order estimation. Second, they found a larger positive amplitude for dot arrays with a numerical ratio of 0.6 as compared to dot arrays with a numerical ratio of 0.3 at medial posterior electrodes between 130–200 ms. Third, the authors observed a larger negative amplitude for descending compared to ascending sequences at left frontal and left parieto-temporal electrodes between 300–600 ms.

Although non-symbolic processing is informative about general aspects of ordinality processing mechanisms, it is unclear if the observed results also translate to ordinal processing of numerals. Indeed, dot arrays elicit different behavioral characteristics as compared to numerals. For example, the reverse distance effect is a well-established finding in numerical order processing, but it is entirely absent when processing the order of dot arrays [17].

Against this background, our study had two primary objectives. First, we sought to identify electrophysiological markers of symbolic numerical ordinality processing using the three-item order verification task. Second, provided that we could find such correlates, we aimed to investigate their associations with individual differences in arithmetic fluency. Beyond these goals, we wanted to explore distance effects of ordinality processing related to our ERP findings. We also set out to reproduce behavioral findings related to the order verification task, that is, a reverse distance effect for ordered sequences with distances of one and a canonical distance effect for unordered sequences for both response times and solution accuracies.

For this purpose, we recorded EEG signals from a group of healthy adults solving a three-item symbolic numerical order verification task. This enabled us to identify ERP components associated with central characteristics of symbolic ordinal number processing. Based on related prior literature, we expected to find differences between ordered and unordered sequences as well as interactions with numerical distance.

Prior studies differed substantially from our design. In particular, Turconi et al. [18] used a two-item comparison task and Rubinsten et al. [31] used non-symbolic dot arrays, whereas

our study employed a three-item symbolic order verification task. Therefore, we did not have specific hypotheses regarding timing, electrode locations, or the sign/direction of potential ERP mean amplitude differences. For this reason, we decided to pursue a data-driven exploratory approach, which means that we did not select time segments or channel locations a priori. Instead, we derived these parameters from peaks in the grand average ERP waveforms aggregated over all conditions and participants [32]. We also collected data from a speeded paper-pencil arithmetic fluency task to test potential relationships of ERP markers with individual arithmetic skills.

## Methods

### Sample

Our sample consisted of 73 right-handed participants (44 female and 29 male) aged between 18 and 39 years (mean age 22.3 with a standard deviation of 3.3 years). Our exclusion criteria, which we defined prior to data collection, comprised left-handedness, age less than 18 years or greater than 40 years, neurological or psychiatric disorders, and people in high-risk groups for COVID-19. Originally, we obtained data from 80 participants, but we had to exclude seven recordings prior to data analysis for the following reasons: five datasets had severe data quality issues (extensive movement and/or large technical artifacts), one participant had a brain lesion, and one participant had dyslexia.

Recruitment for this study took place from December 18, 2020, to April 30, 2021. All participants gave prior written informed consent. We offered psychology students the option to receive course credit for their participation. The Ethics Committee of the University of Graz approved the whole study, and we performed all procedures in accordance with relevant guidelines and regulations.

### Procedure

The entire experimental session lasted for about two hours and consisted of several blocks. First, we informed participants about the motivation and main goals of our study, after which they gave written informed consent. Next, we collected sociodemographic data (sex, age, whether they had diagnosed neurological or psychiatric disorders and/or diagnosed learning disabilities, and if they were part of a high-risk group for COVID-19). After that, participants completed the short version of the Edinburgh Handedness Inventory [33]. Finally, they took part in two experimental blocks: a paper-pencil task to assess arithmetic fluency as well as a computerized order verification task [9]. We presented these blocks in pseudo-randomized order (i.e., half of our sample started with the arithmetic fluency block, whereas the other half started with the order verification block).

Before starting the order verification task, we donned the head cap and active EEG electrodes, since we recorded EEG only during this part of the experiment. Before this experimental block started, participants worked on five practice trials to get familiar with the task, and they took a short break (approximately three minutes) after the first half of the EEG block. We instructed participants to respond as fast and as accurately as possible with their right index and middle fingers, which rested on one of the two keys of a response pad. We counterbalanced the assignment of key/answer mappings by instructing half of the participants to use their index finger for "ordered" and middle finger for "unordered" responses, while the other half used their middle finger for "ordered" and index finger for "unordered" responses. In the next sections, we describe our instruments and stimulus material in more detail.

## Psychometric tests and instruments

We used a German translation of the Edinburgh Handedness Inventory [33]. Participants had to indicate their dominant hand for ten different common tasks (e.g., using a toothbrush) by checking the appropriate column (left or right). If their preference for a hand was very strong, they could assign two checks. If they were indifferent, they checked both columns. We summed all checks for each column, yielding column sums $L$ and $R$. The laterality quotient, calculated as $(R - L) / (R + L)$, distinguishes between left-handedness (a score less than −40%), right-handedness (a score greater than 40%), and ambidexterity (a score between −40% and 40%).

The arithmetic fluency paper-pencil task [9] measures how quickly individuals solve arithmetic problems within a given time limit. The first subtest consists of easy problems, namely 64 multiplications (e.g., $5 \times 7$), 128 additions (e.g., $4 + 7$), and 128 subtractions (e.g., $16 − 8$), with a time limit of 90 seconds per operation. The second subtest consists of more complex problems, namely 60 multiplications (e.g., $39 \times 5$), 60 additions (e.g., $30 + 98 + 59$), and 60 subtractions (e.g., $82 − 31$), with a time limit of 120 seconds per operation. The total number of correctly solved problems equals the individual arithmetic fluency score.

## Stimulus material and trial definition

Each stimulus item of the computerized order verification task consisted of a sequence of three Arabic numerals from 1 to 9. Participants had to decide whether the presented sequence was ordered or not. In addition, we varied the following properties across items: the sequence could be numerically ordered or unordered; the distance (difference) between adjacent numbers could be either 1 or 2; and the sequence could be ascending or descending (only the first two numerals categorized unordered sequences as ascending or descending).

After excluding sequences where the order can be determined from only the first two numerals instead of all three (such as 1–2–3 and 9–8–7), the final stimulus set consisted of 40 distinct items. We presented each unique item repeatedly (either 10, 12, or 20 times, depending on the condition), resulting in 60 sequences per condition and 480 sequences in total. For example, there are six unique sequences in the ordered/distance 1/ascending condition, so we presented each of these sequences 10 times. Similarly, there are five unique ordered/distance 2/descending sequences and only three unique unordered/distance 2/descending sequences, so we presented each sequence 12 and 20 times, respectively (S1 Table lists the complete stimulus set). We illustrate this $2 \times 2 \times 2$ design with an example sequence for each combination in Table 1. Note that we did not analyze direction (ascending vs. descending) in this study, so we collapsed these two levels into a $2 \times 2$ design with factors "order" and "distance".

We used PsychoPy [34] for stimulus presentation (black text on white background) as well as for recording response times. Fig 1 illustrates the timing of a trial in the main experiment.

**Table 1. Examples of sequences for each combination of factor levels.**

| Sequence | Order | Distance | Direction |
| --- | --- | --- | --- |
| 2–3–4 | ordered | 1 | ascending |
| 3–2–1 | ordered | 1 | descending |
| 1–3–5 | ordered | 2 | ascending |
| 5–3–1 | ordered | 2 | descending |
| 2–3–1 | unordered | 1 | ascending |
| 3–2–4 | unordered | 1 | descending |
| 3–5–1 | unordered | 2 | ascending |
| 5–3–7 | unordered | 2 | descending |

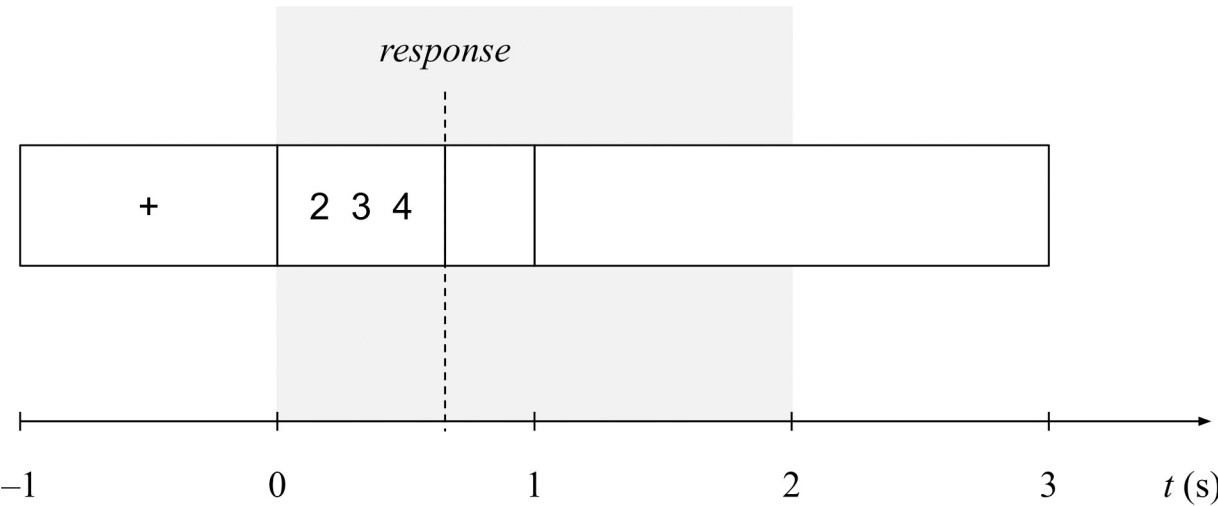

**Fig 1. Timing of a trial.** The dotted vertical line indicates a response, after which the stimulus disappeared. Participants had a maximum of two seconds to respond with a keypress (gray background). The sequence stayed on the screen for a maximum of one second.

First, a fixation cross appeared for one second, followed by a sequence of three single-digit numbers. The stimulus disappeared after one second or the participant's response, whichever occurred earlier. We recorded responses up to two seconds after stimulus onset. A trial ended with one additional second of a blank screen. The total length of each trial was therefore four seconds. We recorded response times as well as correctness of each trial. When analyzing average response times, we only considered correctly solved trials.

## Recording setup

Participants sat in a comfortable office chair in front of a 24-inch flat screen monitor with a resolution of 1920 × 1080 and a refresh rate of 120 Hz. They pressed one out of two keys on a Cedrus RB-740 response pad to indicate whether the presented sequence was ordered or not. In addition to measuring response times, we used a Biosemi ActiveTwo amplifier to record the EEG with 64 active electrodes arranged in the extended 10–20 system. Reference and ground electrodes were placed to the left and right of channel POz, respectively. The amplifier also recorded response pad events through its trigger interface. We recorded all signals with a sampling frequency of 512 Hz, and for EEG channels we set a low-pass edge frequency of 104 Hz and no high-pass filter.

## EEG analysis

We used MNE-Python [35] to analyze EEG data in all processing stages. First, we inspected time series and power spectral densities across all 64 channels to manually identify and remove electrodes containing an excessive amount of noise, large artifacts, or no signals of interest (due to weak contact with the scalp surface). We then re-referenced the remaining channels to their average activity. After that, we interpolated missing channels using spherical splines.

Next, we visually inspected EEG signals within epochs defined as 250 ms before and 750 ms after stimulus onset. We marked all segments containing artifacts such as muscle activity or movement artifacts. In our subsequent analyses, MNE-Python automatically excluded all epochs containing artifact annotations.

Finally, we used Independent Component Analysis (ICA) to remove ocular artifacts such as eye blinks and eye movement [36]. In particular, we first applied a high-pass filter at 1 Hz,

performed ICA based on time points within epochs (but excluding manually marked artifact segments), and manually selected components representing ocular activity based on topographic mappings, power spectral densities, and time courses. We identified between one and three ocular components for each participant, which we set to zero. Lastly, we back-projected the remaining components, which resulted in clean, artifact-free EEG signals.

Based on the preprocessed clean EEG data, we first segmented the continuous signals into epochs (as previously mentioned, an epoch started 250 ms before and ended 750 ms after stimulus onset). We used the 250 ms time window before stimulus onset as our baseline. Because we had no strong prior hypotheses on the location and/or timing of relevant ERP components, we implemented a collapsed localizer approach [32]. This means that we selected segments around peaks based on the grand average waveform over all conditions and participants, separately for each channel.

After defining time segments of interest, we proceeded with computing channel-wise average waveforms for each condition of interest: ordered sequences, unordered sequences, ordered sequences with a distance of one, ordered sequences with a distance of two, unordered sequences with a distance of one, and unordered sequences with a distance of two. In all conditions, we included only correctly solved trials. This procedure resulted in six groups of channel-specific averages (ERPs) per participant for the following three contrasts: ordered vs. unordered sequences, distance 1 vs. distance 2 within ordered sequences, and distance 1 vs. distance 2 within unordered sequences.

## Statistical analysis

We analyzed group-level contrasts for response times, logarithmic error rates (raw error rates are typically right-skewed, so we applied a log transformation), and mean ERP amplitudes for previously defined time windows. We used repeated measures analyses of variance (ANOVA) with factors "order" and "distance" to statistically assess differences in group means. If the models revealed significant effects, we followed up with Bonferroni-adjusted pairwise $t$-tests.

Since we had no prior hypotheses regarding discriminative channel locations or time segments, we decided to explore ERP results in a data-driven manner. Specifically, our goal was to identify electrode locations as well as time segments with the largest differences for each of the three contrasts. To assist with this selection, we created topographic plots by mapping average ERP differences within a specific time segment across all channels. That way, we picked between two and three electrodes at the center of clusters with the largest difference values. We also took magnitudes of differences into account and selected only electrodes showing meaningful activities. After identifying electrodes and time segments, we performed one-sample $t$-tests (one for each electrode in a particular time segment) to determine if a difference deviated significantly from zero or not.

Finally, we correlated individual arithmetic fluency scores with behavioral measures (response time and error rate) as well as with average ERP amplitudes in the previously identified channel locations and time segments.

For statistical analyses, we used R with packages afex (repeated measures ANOVA), effect-size (partial $\eta^2$, or $\eta_p^2$), and emmeans (marginal means and pairwise post-hoc tests) in addition to dplyr, ggbeeswarm, ggplot2, glue, patchwork, purrr, readr, stringr, and tidyr for data wrangling and plotting.

## Open data policy

Our analyses are fully reproducible, because all data sets and analysis scripts are publicly available on OSF (osf.io/7udzb). The repository contains detailed instructions on how to run the scripts.

## Results

### Behavioral data

**Response time.** We only used correctly solved trials to derive response time results. Furthermore, we found no differences between the two counterbalanced key/answer mapping groups, so we will ignore this factor in all subsequent analyses (see S2 and S3 Tables). Fig 2 illustrates mean response times for ordered sequences, which were 0.98 s (for both distance 1 and distance 2), as well as for unordered sequences, which were 1.14 s (distance 1) and 1.04 s (distance 2).

We analyzed response time data with repeated measures ANOVA using within-subject factors "order" and "distance". The model yielded significant main effects of "order" ($F(1, 72) = 193.64$, $\eta_p^2 = .73$, $p < .001$) and "distance" ($F(1, 72) = 98.84$, $\eta_p^2 = .58$, $p < .001$) as well as a significant interaction ($F(1, 72) = 188.53$, $\eta_p^2 = .72$, $p < .001$). A paired $t$-test on estimated

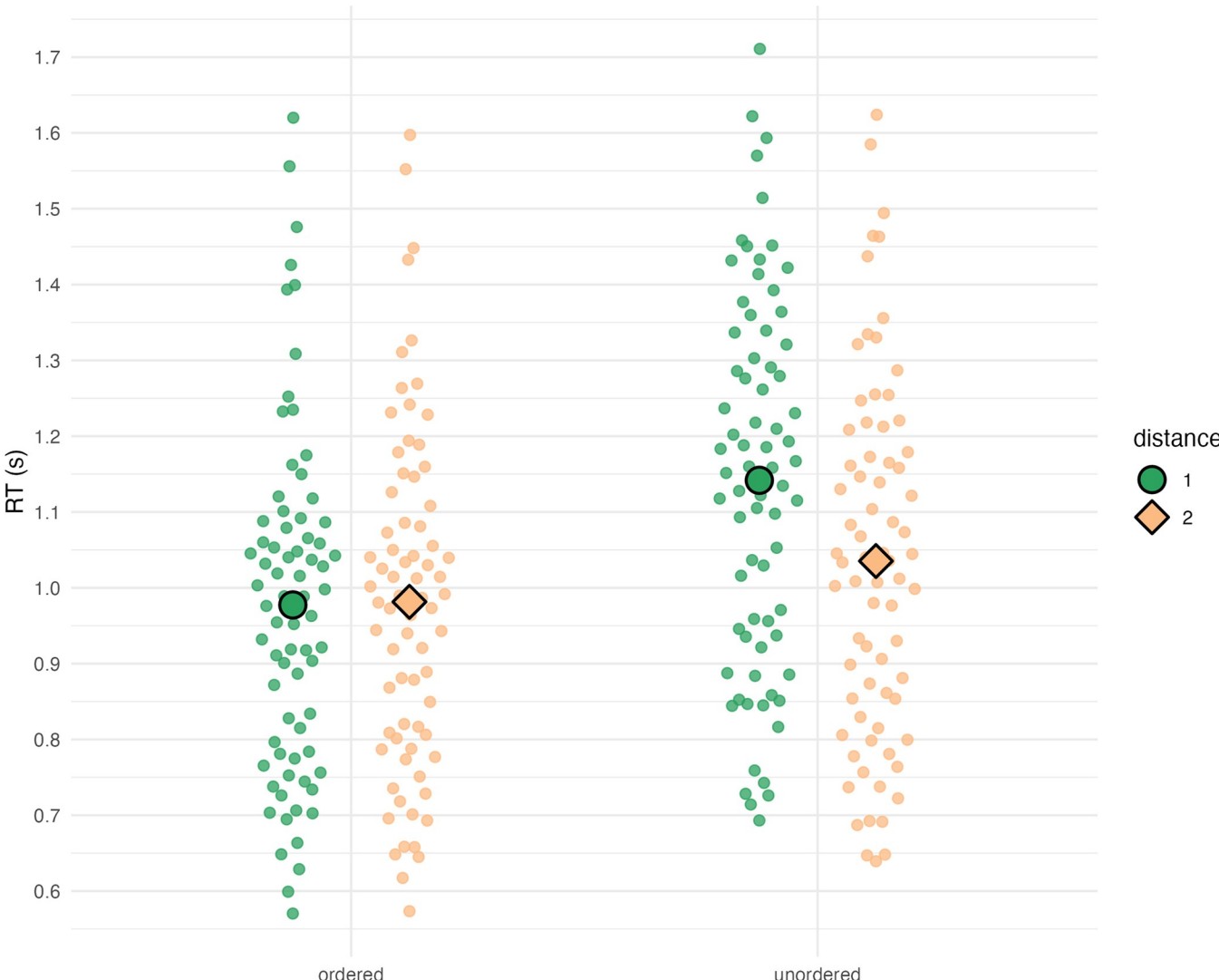

**Fig 2. Average response times.** From left to right: 0.98 s (ordered distance 1), 0.98 s (ordered distance 2), 1.14 s (unordered distance 1), and 1.04 s (unordered distance 2). Large green dots indicate response times for distance 1 sequences, whereas large orange diamonds indicate response times for distance 2 sequences. Smaller points indicate mean response times for individual participants.

marginal means of ordered vs. unordered confirmed our initial expectation that ordered trials are solved faster than unordered trials (with a difference of −0.109 s, $t(72) = -13.915$, $p <$ .001). To further investigate the significant interaction, we conducted Bonferroni-adjusted pairwise $t$-tests on estimated marginal means as defined by the interaction. These follow-up tests yielded significant differences ($p < .001$) between all pairs except for ordered distance 1 vs. ordered distance 2. In particular, we did not find the expected reverse distance effect within ordered sequences (with a difference of −0.004 s, $t(72) = -0.609$, $p = 1$), whereas the canonical distance effect was present within unordered sequences (with a difference of 0.107 s, $t(72) =$ 16.675, $p < .001$).

**Error rate.** We excluded missing responses and used only correct and incorrect responses when calculating error rates. Even though four participants had a relatively high percentage of missing responses (46%, 30%, 26%, and 22%), we decided to retain these datasets for subsequent analysis. The reason for this choice is that error rates were less than or equal to 10%, making them comparable to those of other participants.

Overall error rates varied between 1% and 34% between individuals. Fig 3 illustrates mean error rates for ordered sequences, which were 7.4% (distance 1) and 9.8% (distance 2), as well as for unordered sequences, which were 11.2% (distance 1) and 4.7% (distance 2).

A repeated measures ANOVA with logarithmic error rate as the dependent variable and within-subject factors "order" and "distance" yielded significant main effects of "order" ($F(1,$ 72) = 18.41, $\eta_p^2 = .20$, $p < .001$) and "distance" ($F(1, 72) = 32.81$, $\eta_p^2 = .31$, $p < .001$) as well as a significant interaction ($F(1, 72) = 60.96$, $\eta_p^2 = .46$, $p < .001$). A paired $t$-test on estimated marginal means of ordered vs. unordered yielded a significant, albeit small, difference of 1.51% ($t(72) = 4.29$, $p < .001$). This means that ordered sequences were associated with slightly higher error rates than unordered sequences, which is not in line with our previous expectations. Bonferroni-adjusted pairwise $t$-tests on marginal means as defined by the interaction yielded significant differences ($p < .001$) between all pairs except for ordered distance 2 vs. unordered distance 1 sequences. This means that, in contrast to response times, we found both distance effects in error rates. Within ordered sequences, distance 1 sequences were associated with a lower error rate than distance 2 sequences (with a difference of −2.4%, $t(72) = -3.612$, $p < .01$), which is the expected reverse distance effect. We also found the canonical distance effect within unordered sequences, meaning that distance 1 sequences showed higher error rates than distance 2 sequences (with a difference of 6.5%, $t(72) = 7.433$, $p < .001$).

## ERP data

**Preprocessing.** On average, we interpolated 5.3 ± 2.6 (mean ± standard deviation) bad channels per participant (individual channel selections are available in the data repository). Furthermore, we dropped 12.2 ± 12.9 (mean ± standard deviation) epochs (from a total of 480) per participant.

**Collapsed localizer.** Fig 4 shows the grand average waveform computed across all participants and conditions for one representative channel PO8 (with clearly visible peaks). We used this average waveform to define the following time segments around positive and negative peaks: 85–135 ms, 140–190 ms, 190–275 ms, 285–475 ms, and 500–750 ms.

**Ordered vs. unordered.** We found the largest differences between waveforms of ordered and unordered sequences in the last time segment (500–750 ms). Specifically, there were three distinct clusters with differences greater than 0.5 µV at C2, AF7, and P9, respectively. All other time segments did not contain clusters of large differences. Fig 5 illustrates these findings (see also S1–S3 Figs for additional visualizations of these differences).

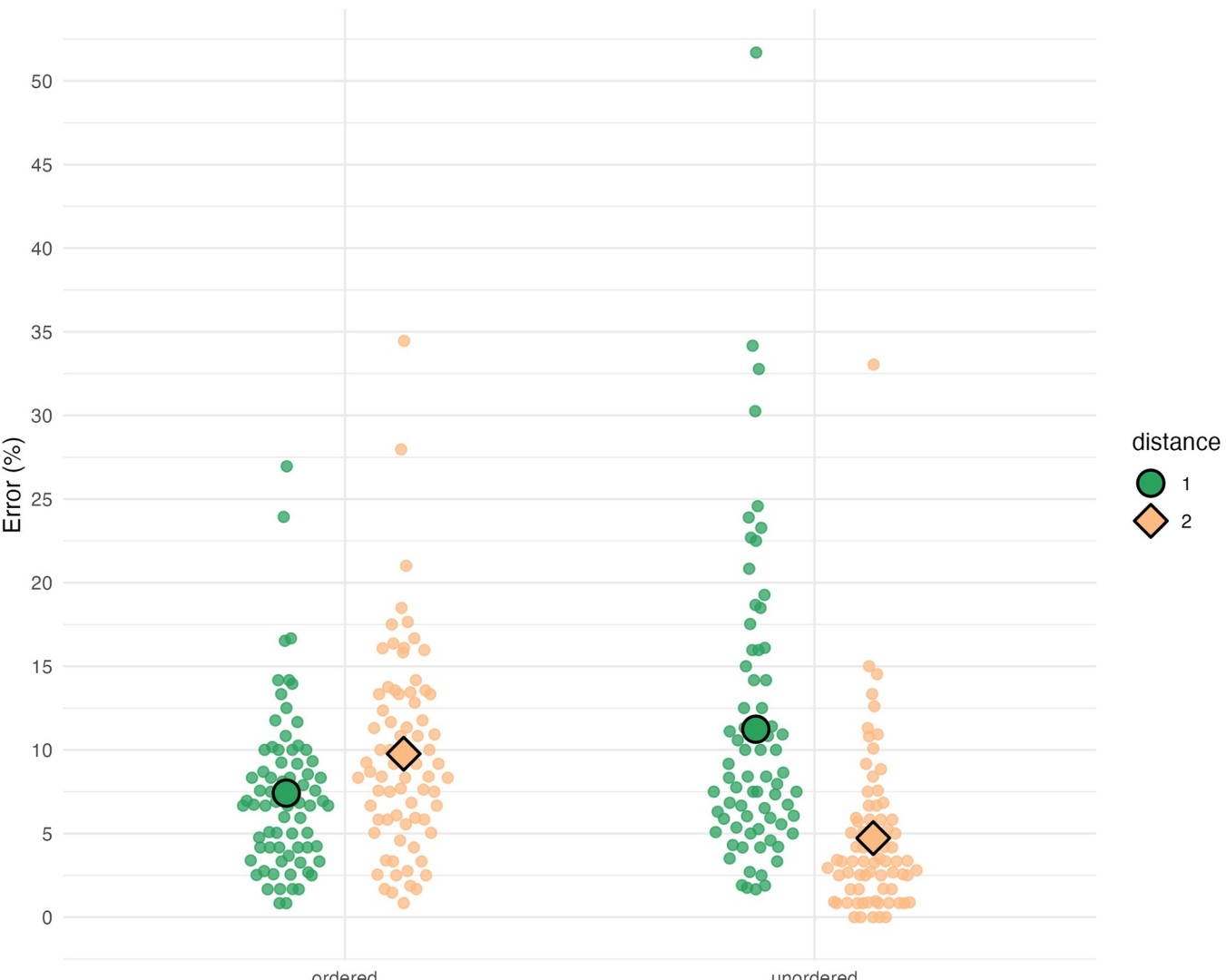

**Fig 3. Error rates.** From left to right: 7.4% (ordered distance 1), 9.8% (ordered distance 2), 11.2% (unordered distance 1), and 4.7% (unordered distance 2). Large green dots indicate mean error rates for distance 1 sequences, whereas large orange diamonds indicate mean error rates for distance 2 sequences. Smaller points represent error rates for individual participants.

We conducted one-sample $t$-tests for each of the three channels to determine if the observed amplitude differences between ordered and unordered sequences differed significantly from zero. Indeed, differences at all three electrode sites were statistically different from zero, with a difference of 0.53 μV at C2 ($t(72) = 4.84$, $p < .001$), −0.80 μV at AF7 ($t(72) = -4.19$, $p < .001$), and −0.83 μV at P9 ($t(72) = -4.52$, $p < .001$).

**Ordered distance 1 vs. distance 2.** Comparing distance 1 to distance 2 within ordered sequences yielded weaker and more diffuse clusters. We identified the third segment (190–275 ms) to contain the largest differences at electrodes Fz and PO8 (see Fig 6 and also S4 Fig).

One-sample $t$-tests for each site resulted in a significant difference of 0.70 μV at PO8 ($t(72) = 5.77$, $p < .001$) as well as a significant difference of −0.30 μV at Fz ($t(72) = -2.61$, $p < .05$).

**Unordered distance 1 vs. distance 2.** The same distance comparison within unordered sequences revealed the largest differences within the last segment (500–750 ms). We identified two weak and diffuse clusters at electrodes AFz and F8 (see Fig 7 and also S5 Fig).

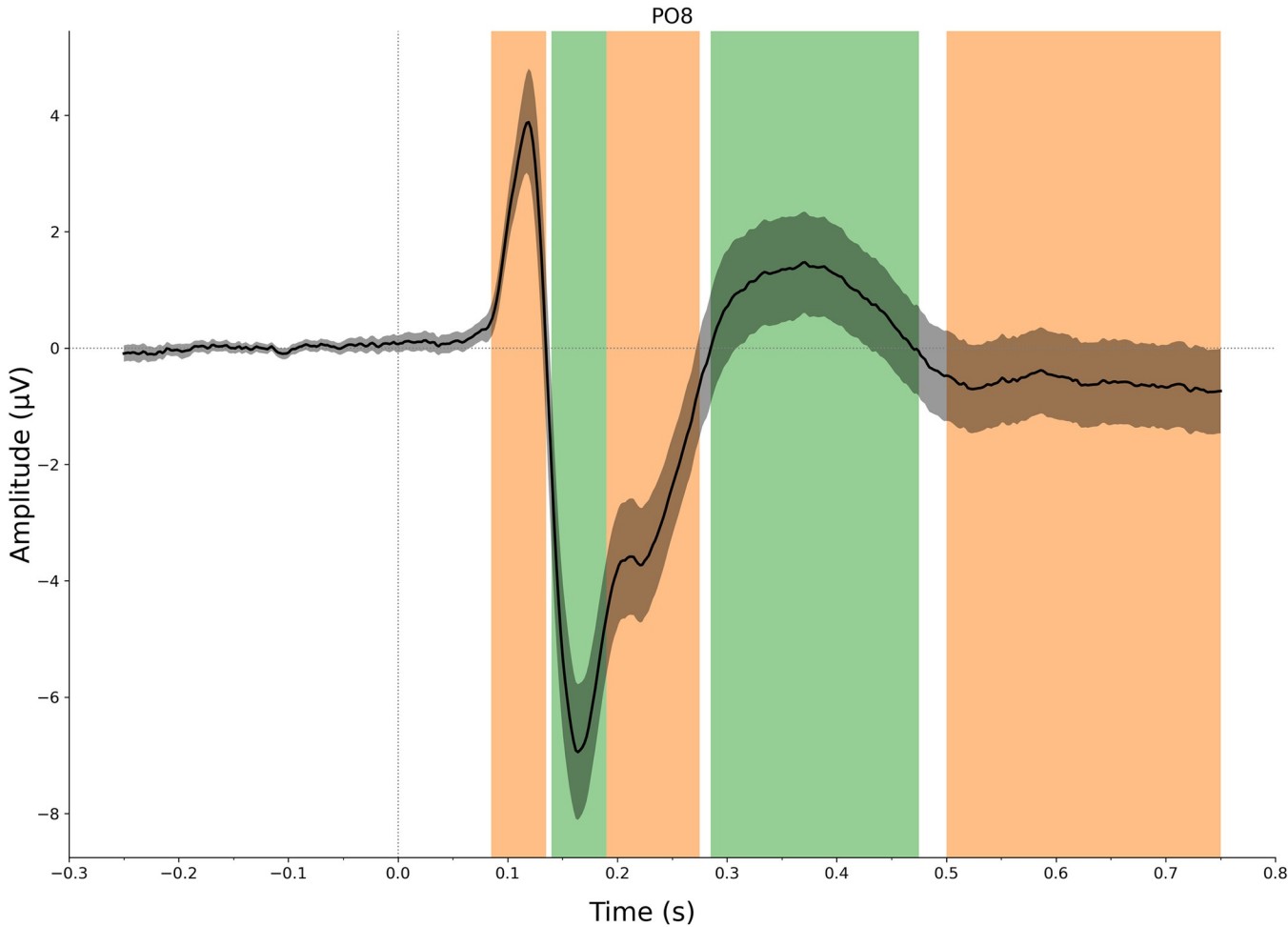

**Fig 4. Collapsed localizer on channel PO8.** The black line corresponds to the grand average with the gray ribbon indicating its 95% confidence interval. Based on this waveform, we identified the following segments: 85–135 ms, 140–190 ms, 190–275 ms, 285–475 ms, and 500–750 ms.

The mean difference at F8 was 0.94 μV, which was statistically significant ($t(72) = 4.77$, $p < .001$). The difference was smaller, albeit statistically significant, at AFz with −0.50 μV ($t(72) = −2.52$, $p < .05$).

## Psychometric data

Arithmetic fluency scores ranged from 98 to 347 with a mean of 194 and standard deviation of 55.2. Fluency correlated significantly with mean response time ($r = −.445$, $t(71) = −4.189$, $p < .001$), but we found no significant correlation with error rate ($r = −.185$, $t(71) = −1.584$, $p = .118$).

We also correlated arithmetic fluency with behavioral distance effects within ordered sequences, but neither response times ($r = −.115$, $t(71) = −0.975$, $p = .333$) nor error rates ($r = .020$, $t(71) = 0.169$, $p = .866$) resulted in significant correlations. Distance-related response time differences within unordered sequences correlated significantly with arithmetic fluency ($r = −.249$, $t(71) = −2.168$, $p < .05$), whereas there was no significant correlation for corresponding error rates ($r = −.136$, $t(71) = −1.155$, $p = .252$).

Furthermore, we computed Pearson correlation coefficients of arithmetic fluency versus mean amplitude differences (ordered vs. unordered) at previously identified electrode

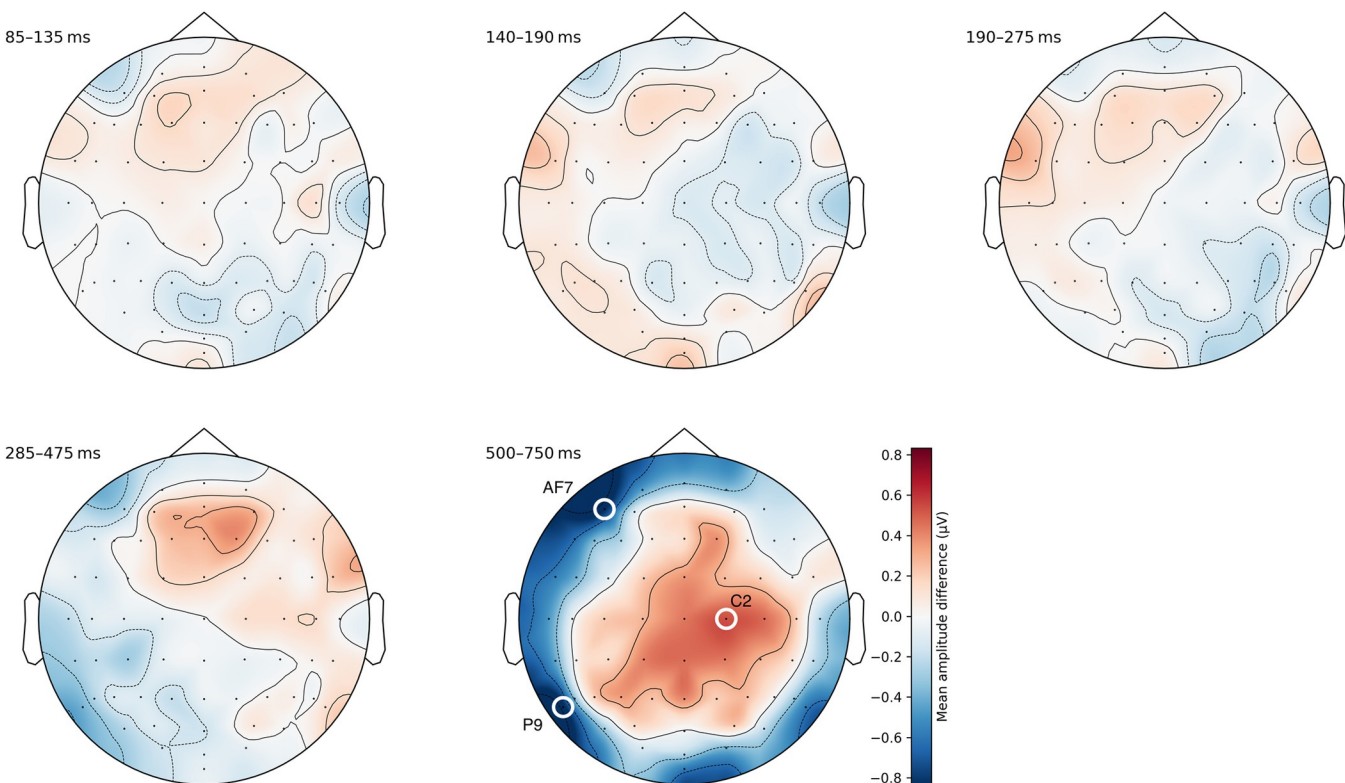

**Fig 5. Topographic difference maps of average amplitudes between ordered and unordered sequences.** The first four time segments showed only small differences, whereas the last time segment between 500–750 ms contains three distinct clusters of meaningful differences around C2, AF7, and P9.

locations C2, AF7, and P9 for the time segment between 500–750 ms. Whereas correlations at P9 ($r = -.159$, $t(71) = -1.355$, $p = .180$) and C2 ($r = .221$, $t(71) = 1.908$, $p = .060$) were not significant, we found a significant correlation at AF7 ($r = -.282$, $t(71) = -2.475$, $p < .05$). We did not find any significant correlations of the distance effects (mean amplitudes at previously defined electrode locations and time windows) in either ordered or unordered sequences with arithmetic fluency.

## Discussion

Our study has revealed, for the first time, specific ERP components associated with the three-item symbolic numerical order verification task, demonstrating their sensitivity to ordinality. We will expound upon potential interpretations and limitations of this finding, but we will start with the behavioral results (response times and error rates).

We could replicate established behavioral findings, namely both the canonical and reverse distance effects. Several previous studies have associated the reverse distance effect with ordinality processing. Specifically, participants are faster and more accurate in recognizing sequences with consecutive numbers (such as 3–4–5) than sequences with inter-item distances of two (such as 3–5–7) as ordered [3]. However, we observed such a reverse distance effect only in error rates. That is, participants performed more accurately for an inter-item distance of 1 as compared to 2 (with error rates of 7.4% and 9.8%, respectively), but there was no significant difference in response times between the two distances (0.98 s in both conditions). This absence is in line with some previous findings, which failed to observe the reverse distance effect at either the group level or in certain individuals [25, 37]. One reason for the absence of

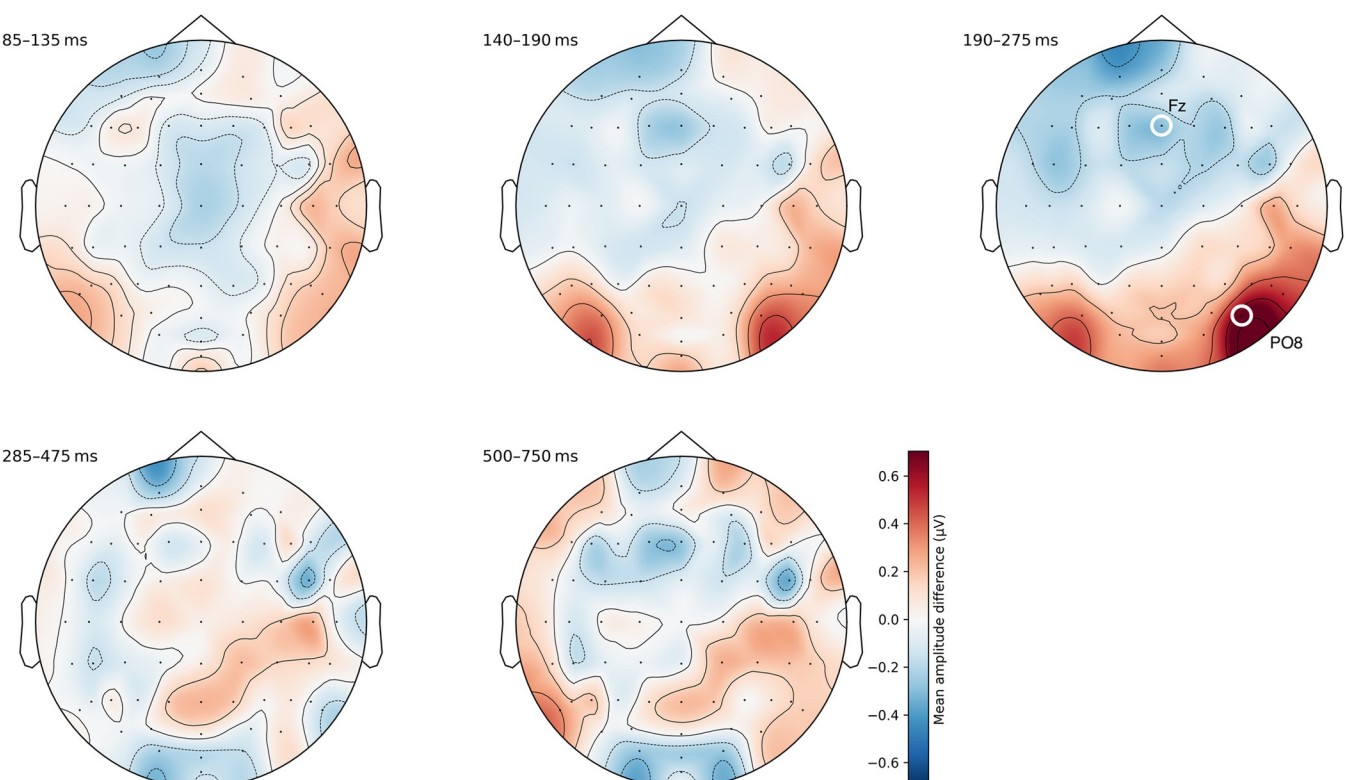

**Fig 6. Topographic difference maps of average amplitudes between distance 1 and distance 2 within ordered sequences.** The third time segment between 190–275 ms showed the largest differences in two distinct clusters around Fz and PO8.

this effect could be variability on the individual level. In their sample of 397 participants, Vogel et al. [25] found that 43% showed a reverse distance effect, 5% showed evidence for an opposite effect (i.e., a canonical distance effect), and 52% did not exhibit any distance effect at all. Variation in group compositions might therefore explain the overall presence or absence of the reverse distance effect. Another explanation could be that our stimulus set did not include the sequence 1–2–3, which is associated with the smallest processing time among ascending consecutive sequences [27]. Therefore, the difference between distance 1 and distance 2 sequences tends to become less pronounced and can even be completely absent (not significant) [37].

In line with previous findings, we found a canonical distance effect in unordered sequences in both response times and error rates. In our study, participants recognized unordered distance 1 sequences more slowly and less accurately (1.14 s and 11.2% errors) than distance 2 sequences (1.04 s and 4.7% errors). In addition, participants recognized ordered sequences faster than unordered sequences. Assuming that they used more efficient strategies, such as memory retrieval, for consecutive ordered sequences, it appears that they tended to rely on slower magnitude comparisons for unordered sequences [3, 5, 25], which explains the average difference in response times between ordered and unordered sequences. In this context, it is worth mentioning that somewhat unexpectedly, participants recognized unordered sequences more accurately than ordered sequences (8.0% vs. 8.6%). However, this is a small difference, driven mostly by the low error rate in unordered distance 2 trials (and in any case, the main effect of order is superseded by the significant interaction with distance).

We will now discuss the ERP correlates associated with the three central features of symbolic numeric order processing. First, when comparing ERP waveforms of ordered vs.

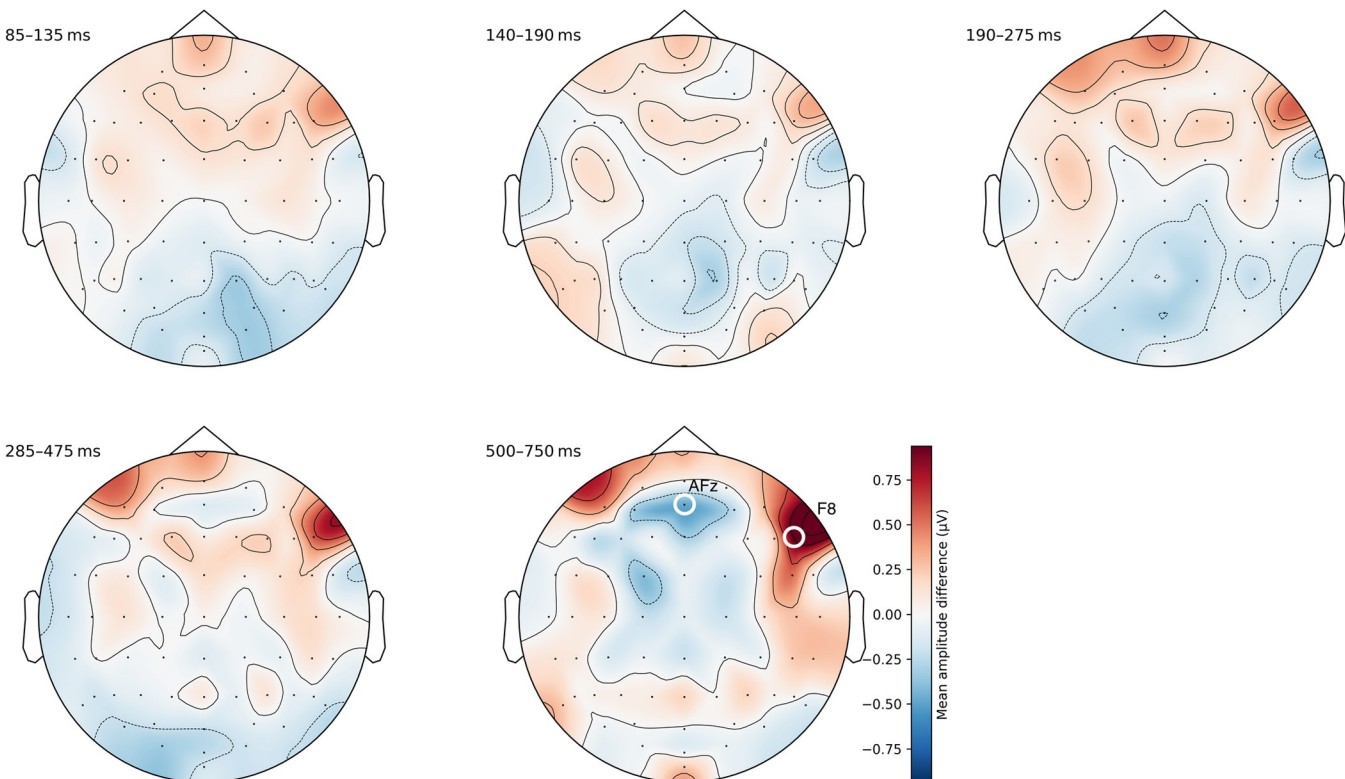

**Fig 7. Topographic difference maps of average amplitudes between distance 1 and distance 2 within unordered sequences.** The last time segment between 500–750 ms showed the largest differences in two clusters around AFz and F8.

unordered sequences, we found a significant difference in the time window of 500–750 ms at three distinct electrode clusters centered around C2, AF7, and P9. At C2, the mean amplitude for ordered sequences was 0.53 μV higher than for unordered sequences, whereas it was 0.80 μV and 0.83 μV lower at AF7 and P9, respectively (see also S2 and S3 Figs). As far as we are aware, our study is the first to investigate ERP correlates related to symbolic ordinality processing using numerals, making direct comparisons with prior research challenging. For example, Rubinsten et al. [31] used non-symbolic dot array triplets and found an effect of order as early as 80–130 ms at right parietal and lateral occipital scalp electrodes. Although such early effects are usually associated with differences in stimulus parameters [38], the authors argue that this difference could also be related to automatic non-symbolic order estimation. Our study revealed no such differences in early time windows around P1 (85–135 ms) or N1 (140–190 ms), indicating that stimuli did not differ across conditions in terms of visual properties such as luminance, size, spatial frequencies, and so on. While it is plausible that ordered and unordered sequences in dot arrays can be differentiated early and potentially also implicitly in the processing stream [39], this might not be the case for symbolic stimuli. In contrast, our results indicate that accessing the meaning of ordered numerals occurs at a later time window and potentially engages different brain regions.

The second major result related to our ERP findings pertains to the reverse distance effect observed in ordered sequences. We found a significant, albeit small, effect of distance (i.e., distance one vs. distance two) in mean amplitude difference. This difference emerged 190–275 ms post-stimulus at two electrode clusters centered around frontal (Fz) and right parieto-occipital (PO8) regions. So far, only one previous EEG study by Turconi and colleagues [18]

has found a distance-related effect for ordered numerals. In particular, the authors reported a distance effect over left parietal electrodes within a time window of 170–210 ms for magnitude comparison, whereas the effect occurred bilaterally at parietal sites between 210–240 ms in the ordinal condition. In contrast to our work, their study used a different task consisting of only two numerals, which might confound the interpretation of their ordinality judgment condition. Indeed, participants had to decide if the presented number is smaller or larger than 15 in their magnitude condition, whereas they had to report if the presented number comes before or after 15 in their ordinality condition. This ordinality task differs significantly from the three-item order verification task, which we employed in our study, and it is not unlikely that participants solved both conditions via magnitude comparisons. In fact, Turconi et al. [18] found a canonical distance effect in behavioral data in both conditions, which provides support for this interpretation.

It is worth noting that the effect we found in our study occurred at an earlier time window than the previously discussed ordinality effect. In addition, its activation pattern is topographically quite different between the reverse and the canonical distance effects. In fact, the pattern of the reverse distance effect is rather focal compared to a more distributed pattern in the canonical distance effect (which we will discuss in the next paragraph). These dissimilarities point to differences in the way ordered and unordered sequences are processed. They are in line with previous findings from fMRI research, which has shown differential brain activation patterns between the two distance effects (e.g., [16]). Additionally, these findings are also compatible with behavioral studies that have suggested a fast retrieval of ordered consecutive items compared to ordered items with a numerical distance of two. For instance, Vogel et al. [10] showed that ordered consecutive items–in particular ascending sequences–facilitate non-numerical decisions (i.e., size judgements) in which the numbers are not relevant for the task (i.e., ordinal Stroop condition). Overall, we argue that the focal EEG correlates at this earlier time window are therefore consistent with the explanation of an automatic and efficient processing of ordinal sequences.

Finally, the third ERP finding relates to the canonical distance effect in unordered sequences. We found a significant effect of distance within unordered sequences at 500–750 ms post-stimulus at two electrode clusters centered around anterior-frontal (AFz) and right frontal (F8) regions. We are not aware of prior neuroimaging research (neither EEG nor fMRI) that has investigated the canonical distance effect within unordered sequences. This makes a direct comparison with existing findings challenging again. However, there are numerous functional neuroimaging studies that have investigated numerical magnitude processing within different experimental settings (e.g., number comparison, number adaptation, and ordinal processing tasks). These findings have provided converging and convincing evidence that brain activation patterns in the prefrontal and parietal lobe can be linked to the mental processing and manipulation (e.g., in the context of calculations) of numerical magnitudes (for reviews, see Nieder and Dehaene [40] and Vogel and De Smedt [11]). It is, therefore, possible that the frontal ERP effects observed in our work reflect numerical magnitude processing in the context of ordinal decisions [3, 5, 25]. While our findings align with previous ERP studies that have identified frontal and parietal electrodes within different time windows ranging from 120 ms to 430 ms in the context of semantic processing (e.g., [41–43]) and from 350 ms to 900 ms in the context of stimulus and response conflict (e.g., [44]), additional research is required to validate our findings and assumptions.

We will now discuss the relationship of electrophysiological correlates with arithmetic fluency. Our results revealed a significant negative correlation between arithmetic fluency and mean ERP amplitude of ordered versus unordered sequences in a time window of 500–750 ms at AF7. All other correlations of fluency, especially with distance-related measures, did not

significantly differ from zero. This is not entirely unexpected, because the association between arithmetic skills and distance-related measures is currently a topic of ongoing discussions. On the one hand, some studies have observed a significant relationship between the reverse distance effects in ordered sequences [8, 10, 25], but on the other hand, several studies have failed to replicate this association [9, 10, 19, 30]. As with the presence or absence of the reverse distance effect in some individuals, the presence or absence of a correlation with arithmetic fluency might be driven by differences in group composition. For instance, our own work has demonstrated that associations with arithmetic skills are significantly larger when restricting group analyses to individuals who show a significant reverse distance effect [25]. Another explanation draws on the low reliability of behavioral difference score measures [45]. Our knowledge about the association of the canonical distance effect in ordinal verification tasks and arithmetic is even more limited, because, to the best of our knowledge, no experimental work has investigated this potential link. Our data indicate that there might not be a relationship of the canonical distance effect with arithmetic fluency. This is surprising, because a number of behavioral and neuroimaging studies have found significant correlations between numerical magnitude processing (i.e., the canonical distance effect in other tasks) and arithmetic skills (see a meta-analysis of Schneider et al. [46]). In contrast, our results indicate that more general processing mechanisms might relate to arithmetic skills: both the ERP correlates of ordered versus unordered sequences and the overall response times correlated negatively with arithmetic fluency. This finding is in line with some of the results from the neuroimaging literature that has demonstrated an association between domain-general brain regions, such as the semantic control network [7, 12, 15], and arithmetic skills. Our work thus raises the question to what extent domain-general and domain-specific factors of ordinality processing relate to arithmetic skills.

Finally, we would like to discuss some limitations as well as highlight future research directions. First, we adopted an exploratory approach in our ERP analysis due to the lack of related prior studies. This means that we did not have strong prior hypotheses regarding the locations and time windows of ERP components sensitive to ordinality and/or numerical distance. However, we were careful to avoid introducing bias in our analysis by using the collapsed localizer approach [32]. Still, our results need to be interpreted cautiously, and follow-up studies to replicate these findings will be essential.

Second, the nature of the three-item ordinal verification task is debated, and some researchers argue that it might index familiarity with specific number sequences rather than ordinality processing [26]. In particular, consecutive sequences might be more familiar, and therefore recognized faster and more accurately compared to sequences separated by an inter-item distance of two. However, familiarity with specific numbers and their ordinal structure might be defined as ordinal knowledge, which certainly plays a role in ordinality processing. However, an alternative task does not exist yet, and because numerous studies have used the three-item order verification task, we believe it is still interesting to determine any ERP correlates associated with this task. An active ordering task, where participants need to sort a number of items, is popular in behavioral studies, especially with children (e.g., [47]), and might be employed in future EEG studies.

Third, it is worth noting that we defined distance in unordered sequences relative to the first number, whereas distance in ordered sequences applies to both the first and second as well as the second and third numbers. For example, the unordered sequence 5–4–6 has a distance of one, even though the two inter-item differences are one and two, respectively. Similarly, we defined 5–7–3 to have a distance of two, with individual inter-item distances of two and four, respectively. This contrasts with ordered sequences, which have a constant inter-item difference of either one or two, depending on the condition. Despite this distinction, we

maintain that our manipulation of numerical distance within unordered sequences remains appropriate and effective.

Finally, we did not analyze ascending and descending directions separately, because the current design would have yielded an insufficiently small number of repetitions per condition required for averaging. Future work could investigate differences in ascending and descending trials with an appropriate design.

In conclusion, our study revealed ERP correlates sensitive to symbolic numeric order processing as measured by the three-item order verification task. We found differences between ordered and unordered sequences in mean amplitudes at left frontal, central, and left parietal sites in the 500–750 ms time window. In addition, we identified distance effects for both ordered sequences at frontal and right parietal sites (between 190–275 ms), as well as for unordered sequences at frontal locations in a 500–750 ms window. This pattern suggests a fast and efficient processing of ordinal sequences that differs from the processing of unordered sequences. The potential association between arithmetic skills and ordinal processing remains an open question to be answered in future work.

## Supporting information

**S1 Table. Stimulus set.** Note that we presented sequences without dashes in the experiment.
(DOCX)

**S2 Table. Key/answer mapping analysis for RT.** Note that all model terms involving the factor "group" are not significant. The degrees of freedom for the $F$ statistic are 1 and 71 for all terms.
(DOCX)

**S3 Table. Key/answer mapping analysis for ERR.** Note that all model terms involving the factor "group" are not significant. The degrees of freedom for the $F$ statistic are 1 and 71 for all terms.
(DOCX)

**S1 Fig. Mean amplitude differences of ordered vs. unordered sequences for each channel and time segment.** Green indicates negative differences, red indicates positive differences.
(TIF)

**S2 Fig. Average ERP waveforms at selected channel locations for ordered (blue) vs. unordered (orange) sequences.** Ribbons indicate 95% confidence intervals.
(TIF)

**S3 Fig. Average ERP difference waveforms (ordered minus unordered) at selected channel locations.** Ribbons indicate 95% confidence intervals.
(TIF)

**S4 Fig. Mean amplitude differences of distance 1 vs. distance 2 ordered sequences for each channel and time segment.** Green indicates negative differences, red indicates positive differences.
(TIF)

**S5 Fig. Mean amplitude differences of distance 1 vs. distance 2 unordered sequences for each channel and time segment.** Green indicates negative differences, red indicates positive differences.
(TIF)

## Acknowledgments

We thank our academic editor Jérôme Prado and both reviewers (Declan Devlin and an anonymous reviewer) for their valuable and helpful comments.

## Author Contributions

**Conceptualization:** Philip Schadenbauer, Nele Schröder, Stephan E. Vogel.

**Data curation:** Clemens Brunner.

**Formal analysis:** Clemens Brunner.

**Investigation:** Philip Schadenbauer, Nele Schröder.

**Methodology:** Clemens Brunner, Stephan E. Vogel.

**Project administration:** Stephan E. Vogel.

**Resources:** Roland H. Grabner.

**Software:** Clemens Brunner.

**Supervision:** Stephan E. Vogel.

**Validation:** Clemens Brunner.

**Visualization:** Clemens Brunner.

**Writing – original draft:** Clemens Brunner.

**Writing – review & editing:** Clemens Brunner, Roland H. Grabner, Stephan E. Vogel.

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
