## [Decision Letter · Decision Letter 0]

10 Jan 2024

PONE-D-23-37780Electrophysiological correlates of symbolic numerical order processingPLOS ONE

Dear Dr. Brunner,

Thank you for submitting your manuscript to PLOS ONE. I have sent it to two expert reviewers and have now received their comments back. As you can see at the bottom of this email, both reviewers find merit in your manuscript. Both notably highlight that your study is interesting and has the potential to be informative for the field. I do agree. However, the reviewers also raise a number of points that relate to issues with interpretation as well as methodology and data analyses. I would like to give you the opportunity to take into account these comments in a major revision. Therefore, I invite you to submit a revised version of the manuscript that addresses the points raised during the review process.

We look forward to receiving your revised manuscript.

Kind regards,

Jérôme Prado

Academic Editor

PLOS ONE

Journal Requirements:

Reviewers' comments:

Reviewer's Responses to Questions

**Comments to the Author**

1. Is the manuscript technically sound, and do the data support the conclusions?

Reviewer #1: Yes

Reviewer #2: Partly

2. Has the statistical analysis been performed appropriately and rigorously? 

Reviewer #1: Yes

Reviewer #2: Yes

3. Have the authors made all data underlying the findings in their manuscript fully available?

Reviewer #1: Yes

Reviewer #2: Yes

4. Is the manuscript presented in an intelligible fashion and written in standard English?

Reviewer #1: Yes

Reviewer #2: Yes

5. Review Comments to the Author

Reviewer #1: General comment: This manuscript describes an interesting and well-motivated study that represents a valid and valuable contribution to the literature. Furthermore, I found it to be well written and enjoyable to read. I have no major concerns with the manuscript. Below are some very minor comments the authors may consider.

Comment 1: The exploratory approach used here seems appropriate and fully justified. However, I think the exploratory nature of the analysis should be made explicit in the abstract.

Comment 2: I would appreciate more clarity on the number of times each sequence was repeated. For instance, on page 7 (lines 164-166) it is stated that each item was presented “repeatedly” but it is not explicitly stated in the main text how many times each item was repeated.

Although this information is provided in the supplementary materials (Table S1), this also raises some questions. In particular, most sequences were repeated 10 times, but some were repeated 12 times, and some were repeated 20 times. It was not immediately clear to me why this was the case. I assume this is to balance the number of trials between order x distance x direction combinations. However, this is not made explicitly clear, nor is it even mentioned that some combinations have more unique items than others.

Therefore, I think it should be briefly explained in the manuscript i) how many times sequences were repeated and ii) how the authors balanced the number of items/repetitions across the order x distance x direction combinations.

Comment 3: When considering the group-level absence of the RDE in this study (based on RTs), the authors should perhaps also consider that this absence may have resulted from their task not including the sequence 1-2-3.

For example, Vos et al. (2021) proposed that the lack of group-level RDEs in their study may have resulted from their task not including the sequences 1-2-3 and 2-3-4. This is because, from a familiarity account of the RDE, consecutive sequences are processed faster due to being more familiar (and thus more easily retrieved from memory). Therefore, because 1-2-3 and 2-3-4 are arguably two of the most familiar consecutive sequences, their exclusion likely reduced the overall familiarity of their consecutive sequences. Furthermore, Vos et al. (2021) argued that some of the included non-consecutive sequences were perhaps relatively familiar (e.g., 2-4-6). As such, familiarity was arguably now balanced between their consecutive and non-consecutive sequences, thus resulting in comparable response times between the two groups (and thus no RDEs).

Therefore, it is certainly notable that the present study also did not include the sequence 1-2-3 and also did not observe group level RDEs (based on RTs). As such, I think the authors should at least consider this as a possible explanation for the absence of the RDE in the present study.

Even if one does not agree with the familiarity interpretation of the RDE, it is has been demonstrated that 1-2-3 is the fastest processed ascending-consecutive sequence (i.e., Sella et al., 2020). Therefore, regardless of why 1-2-3 is processed so fast, its exclusion from the present task would still be expected to result in overall slower response times for consecutive sequences relative to studies including 1-2-3.

Comment 4: In this context, I also have a very minor point about how the authors describe the reverse distance effect on both page 4 (lines 59-61) and page 17 (lines 364-365). For example:

“Second, ordered sequences with an inter-item distance of one (e.g., 2–3–4) are recognized faster and more accurately than ordered sequences with an inter-item distance of two (e.g., 2–4–6), which is known as the reverse distance effect [3,22,23].”

Although this description is fine, I have an issue with the use of 2-4-6 as an example here since this statement strongly implies that 2-4-6 is processed relatively slowly (at least compared to consecutive sequences). The reason this concerns me is because I am simply not sure this is true. For instance, as noted above, Vos et al. (2021) highlights that 2-4-6 is arguably a relatively familiar sequence and thus (from a familiarity perspective) processing of this sequence would actually be expected to be facilitated.

In fact, in my own (currently unpublished) data, I observed that 2-4-6 is one of the fastest processed sequences, being processed faster than many ascending-consecutive sequences (even including 2-3-4). Therefore, I would suspect that if the authors looked at RTs for individual sequences in the present study, they would likely also observe 2-4-6 to be one of the fastest processed sequences.

In this context, therefore, I think it is best to avoid using 2-4-6 as an example here since it implies that 2-4-6 is processed slowly despite that probably not being the case. Therefore, it may be more appropriate to simply use another example (e.g., 3-4-5 and 3-5-7) instead of 2-3-4 and 2-4-6.

Comment 5: On page 17 (line 371) there is seemingly an error where the citation is given both in numbers “[24, 35]” and in text “(Vogel et al., 2021; Vos et al., 2021)”.

Reviewer #2: In the current study, Brunner and colleagues aimed at investigating the ERP components associated with symbolic numerical order processing. Using a three-digits order verification task, they manipulated both the order (ordered versus unordered) and the inter-item numerical distance (one versus two) of the sequences. Participants had to determine whether the three digits were ordered or not and they also performed an arithmetic test. Both behavioral and EEG results are reported. Correlation analyses were then computed between symbolic numerical order processing and arithmetic fluency measures.

The study is very interesting and well written. However, I have several comments that I feel could improve the manuscript. Overall, I think this study should ultimately be accepted, provided authors can convincingly address the below points.

Introduction:

- Distinction between order and magnitude (cardinality) processing. For me, it is still unclear how order can be extracted independently of magnitude. As the authors mentioned in reference to Turconi et al.'s study, it could be argued that both order and magnitude conditions in that study actually rely on magnitude processing (and the behavioral results for the two tasks are exactly the same): what is really the difference between being "before" or "smaller" than 15 (or after/greater than 15)? While I agree that using multiple numerical sequences solves part of the problem, I still believe that an additional control condition in which, for example, sequences of letters would be determined as ordered or unordered, could help distinguish between numerical/magnitude comparisons and "pure" order processing.

- Related to that, I think that a clear(er) distinction between the cardinal and reverse distance effects should be included in the introduction:

o In line 42, the canonical distance effect is introduced but, as the authors mention, it appears in both ordinal and magnitude conditions (based on the study by Turconi and colleagues). This may be confusing, as the cardinal distance effect is generally considered a marker of magnitude processing; and an inverse distance effect would have been expected for order processing.

o In line 59, the reverse distance effect is introduced but should be clearly distinguished from the cardinal distance effect with regard to their origin.

o In line 68, the canonical distance effect is reintroduced, but in the context of order processing (for unordered sequences).

My suggestion would be to gather all the information about distance effects to clarify what it represents in each context (magnitude comparisons, ordered and unordered sequences).

- In line 66, authors suggest that « individuals process ordinal sequences in qualitatively distinct ways », which qualitatively distinct ways ?

- I agree that the question of whether (and how) order processing and arithmetic skills are associated is relevant. However, I think more emphasis could be placed on why order, in particular, might be a crucial element for arithmetic. For example, it could be mentioned that learning the order of the first number words has been shown to be a necessary step in the acquisition of counting and cardinal knowledge (Wynn, 1992).

Methods:

- It is not mentioned why the authors excluded left-handed participants ?

- How many channels were interpolated ? How many epochs excluded ? Please mention it in the manuscript.

- I understand why the authors decided to analyze their EEG restults in a data-driven manner, but I wonder if this might have biased the results. What if the authors analyzed their results in the same regions or time segments as those defined in Rubinsten and colleagues' study?

- Related to that, why do the authors think that the results for non-symbolic dot arrays would be so different from those for symbolic Arabic numerals with regard to order processing? Could the authors justify why they think order processing is format-specific and not a more general mechanism? A similar question was discussed in a recent study : Wong, B., Bull, R., Ansari, D., Watson, D. M., & Liem, G. A. D. Order processing of number symbols is influenced by direction, but not format. Q. J. Exp. Psychol. 75, 98–117 (2021) : order effect independent of format ».

Results :

- Response time : What is the percentage of correct trials ? Please report it.

- Response time : Please report statistics for the no difference between counterbalanced key/answer mapping groups.

- Unless I missed it, authors did not mention whether they corrected for multiple comparisons in their correlation analyses.

Discussion :

- While I acknowledge the novelty of this study and recognize the potential challenges in directly comparing it to previous research, I recommend incorporating EEG literature related to the processing of Arabic digits. This inclusion would contribute to situating the present findings within the broader context of our understanding of the neurophysiological aspects associated with symbolic numeral coding.

- I feel that the authors simplify the discussion regarding the ERPs and correlations with arithmetic:

o Electrode clusters are very sparse (e.g., C2, AF7 and P9), and the time window selected for ordered versus unordered sequences is quite late compared to previous studies (210-240ms in Turconi et al. ; 130-200ms in Rubinsten et al. ; 500-750ms in this study). The authors should further discuss these points.

o How do the authors interpret the variation in mean amplitude between ordered and unordered sequences, considering that it is sometimes higher (at C2) and sometimes lower (at AF7 and P9) depending on the selected channels? Does the fact that the polarity of the order effect changes depending on the electrode suggest the involvement of different mechanisms in the processing of numerical order? If so, I believe this should be included in the discussion.

o I have the same question about the distance effects, which appear either positive (at PO8 for ordered and F8 for unordered sequences) or negative (at Fz for ordered and AFz for unordered sequences) depending on the electrodes. How do the authors interpret these changes in polarity? Are there any other ERP studies on the magnitude distance effect that reported a comparable contrasting difference at distinct sites?

o How do the authors interpret the significant negative correlation between arithmetic fluency and mean ERP amplitude of ordered versus unordered sequences at AF7 (where mean amplitude for ordered sequences was lower than for unordered sequences) ?

6. PLOS authors have the option to publish the peer review history of their article (what does this mean?). If published, this will include your full peer review and any attached files.

Reviewer #1: **Yes: **Declan Devlin

Reviewer #2: No

---

## [Author Response · Author response to Decision Letter 0]

19 Feb 2024

Please see the separate "Response to Reviewers" document for a formatted version of our responses (which is much easier to read).

We sincerely appreciate all of your valuable comments, which have helped us to substantially improve our manuscript. Therefore, we have added the following sentence to the Acknowledgments section:

We thank our academic editor Jérôme Prado and both reviewers (Declan Devlin and an anonymous reviewer) for their valuable and helpful comments.

Reviewer #1

General comment: This manuscript describes an interesting and well-motivated study that represents a valid and valuable contribution to the literature. Furthermore, I found it to be well written and enjoyable to read. I have no major concerns with the manuscript. Below are some very minor comments the authors may consider.

Thank you for your encouraging and thoughtful review. Please find our responses to each of your comments below.

Comment 1: The exploratory approach used here seems appropriate and fully justified. However, I think the exploratory nature of the analysis should be made explicit in the abstract.

We have changed the following sentence in the abstract accordingly: “To address this gap, we used a three-item symbolic numerical order verification task (with Arabic numerals from 1 to 9) to study event-related potentials (ERPs) in 73 adult participants in an exploratory approach.”

Comment 2: I would appreciate more clarity on the number of times each sequence was repeated. For instance, on page 7 (lines 164-166) it is stated that each item was presented “repeatedly” but it is not explicitly stated in the main text how many times each item was repeated.

Although this information is provided in the supplementary materials (Table S1), this also raises some questions. In particular, most sequences were repeated 10 times, but some were repeated 12 times, and some were repeated 20 times. It was not immediately clear to me why this was the case. I assume this is to balance the number of trials between order x distance x direction combinations. However, this is not made explicitly clear, nor is it even mentioned that some combinations have more unique items than others.

Therefore, I think it should be briefly explained in the manuscript i) how many times sequences were repeated and ii) how the authors balanced the number of items/repetitions across the order x distance x direction combinations.

We have modified the description accordingly: “We presented each unique item repeatedly (either 10, 12, or 20 times, depending on the condition), resulting in 60 sequences per condition and 480 sequences in total. For example, there are six unique sequences in the ordered/distance 1/ascending condition, so we presented each of these sequences 10 times. Similarly, there are five unique ordered/distance 2/descending sequences and only three unique unordered/distance 2/descending sequences, so we presented each sequence 12 and 20 times, respectively (S1 Table lists the complete stimulus set). We illustrate this 2 × 2 × 2 design with an example sequence for each combination in Table 1.”

Comment 3: When considering the group-level absence of the RDE in this study (based on RTs), the authors should perhaps also consider that this absence may have resulted from their task not including the sequence 1-2-3.

For example, Vos et al. (2021) proposed that the lack of group-level RDEs in their study may have resulted from their task not including the sequences 1-2-3 and 2-3-4. This is because, from a familiarity account of the RDE, consecutive sequences are processed faster due to being more familiar (and thus more easily retrieved from memory). Therefore, because 1-2-3 and 2-3-4 are arguably two of the most familiar consecutive sequences, their exclusion likely reduced the overall familiarity of their consecutive sequences. Furthermore, Vos et al. (2021) argued that some of the included non-consecutive sequences were perhaps relatively familiar (e.g., 2-4-6). As such, familiarity was arguably now balanced between their consecutive and non-consecutive sequences, thus resulting in comparable response times between the two groups (and thus no RDEs).

Therefore, it is certainly notable that the present study also did not include the sequence 1-2-3 and also did not observe group level RDEs (based on RTs). As such, I think the authors should at least consider this as a possible explanation for the absence of the RDE in the present study.

Even if one does not agree with the familiarity interpretation of the RDE, it is has been demonstrated that 1-2-3 is the fastest processed ascending-consecutive sequence (i.e., Sella et al., 2020). Therefore, regardless of why 1-2-3 is processed so fast, its exclusion from the present task would still be expected to result in overall slower response times for consecutive sequences relative to studies including 1-2-3.

We have added the following sentences to the paragraph in the discussion where we discuss the absence of the effect: “Another explanation could be that our stimulus set did not include the sequence 1–2–3, which is associated with the smallest processing time among ascending consecutive sequences [27]. Therefore, the difference between distance 1 and distance 2 sequences tends to become less pronounced and can even be completely absent (not significant) [36].”

Comment 4: In this context, I also have a very minor point about how the authors describe the reverse distance effect on both page 4 (lines 59-61) and page 17 (lines 364-365). For example:

“Second, ordered sequences with an inter-item distance of one (e.g., 2–3–4) are recognized faster and more accurately than ordered sequences with an inter-item distance of two (e.g., 2–4–6), which is known as the reverse distance effect [3,22,23].”

Although this description is fine, I have an issue with the use of 2-4-6 as an example here since this statement strongly implies that 2-4-6 is processed relatively slowly (at least compared to consecutive sequences). The reason this concerns me is because I am simply not sure this is true. For instance, as noted above, Vos et al. (2021) highlights that 2-4-6 is arguably a relatively familiar sequence and thus (from a familiarity perspective) processing of this sequence would actually be expected to be facilitated.

In fact, in my own (currently unpublished) data, I observed that 2-4-6 is one of the fastest processed sequences, being processed faster than many ascending-consecutive sequences (even including 2-3-4). Therefore, I would suspect that if the authors looked at RTs for individual sequences in the present study, they would likely also observe 2-4-6 to be one of the fastest processed sequences.

In this context, therefore, I think it is best to avoid using 2-4-6 as an example here since it implies that 2-4-6 is processed slowly despite that probably not being the case. Therefore, it may be more appropriate to simply use another example (e.g., 3-4-5 and 3-5-7) instead of 2-3-4 and 2-4-6.

This is a great argument, and we completely agree that our two examples were not prototypical for the properties we wanted to discuss. Therefore, we have replaced the examples with 3–4–5 and 3–5–7, respectively, as per your suggestion.

Comment 5: On page 17 (line 371) there is seemingly an error where the citation is given both in numbers “[24, 35]” and in text “(Vogel et al., 2021; Vos et al., 2021)”.

This was an oversight; we have removed the text citation.

 

Reviewer #2

In the current study, Brunner and colleagues aimed at investigating the ERP components associated with symbolic numerical order processing. Using a three-digits order verification task, they manipulated both the order (ordered versus unordered) and the inter-item numerical distance (one versus two) of the sequences. Participants had to determine whether the three digits were ordered or not and they also performed an arithmetic test. Both behavioral and EEG results are reported. Correlation analyses were then computed between symbolic numerical order processing and arithmetic fluency measures.

The study is very interesting and well written. However, I have several comments that I feel could improve the manuscript. Overall, I think this study should ultimately be accepted, provided authors can convincingly address the below points.

Thank you for your helpful and constructive comments. We hope that we have adequately addressed your points. Please find our detailed responses below.

Introduction:

- Distinction between order and magnitude (cardinality) processing. For me, it is still unclear how order can be extracted independently of magnitude. As the authors mentioned in reference to Turconi et al.'s study, it could be argued that both order and magnitude conditions in that study actually rely on magnitude processing (and the behavioral results for the two tasks are exactly the same): what is really the difference between being "before" or "smaller" than 15 (or after/greater than 15)? While I agree that using multiple numerical sequences solves part of the problem, I still believe that an additional control condition in which, for example, sequences of letters would be determined as ordered or unordered, could help distinguish between numerical/magnitude comparisons and "pure" order processing.

Several neuroimaging studies have suggested that the neural mechanisms involved in order processing are distinct from those used in magnitude processing (e.g., Lyons et al., 2013). In contrast to comparing two numbers (as used by Turconi et al., 2004), number triplets are designed to better differentiate between magnitude and order processing (but they have other limitations as discussed by Devlin et al., 2022). In addition, existing research indicates that ordinality judgments with letter sequences elicit behavioral (e.g., Vogel et al., 2017) and neuroimaging patterns (e.g., Attout et al., 2022) similar to those observed with number sequences. For these reasons, including a control condition consisting of sequences of letters would be outside the scope of our study.

Attout L, Leroy N, Majerus S. The Neural Representation of Ordinal Information: Domain-Specific or Domain-General? Cerebral Cortex. 2022; 32(6): 1170–1183. https://doi.org/10.1093/cercor/bhab279

- Related to that, I think that a clear(er) distinction between the cardinal and reverse distance effects should be included in the introduction:

o In line 42, the canonical distance effect is introduced but, as the authors mention, it appears in both ordinal and magnitude conditions (based on the study by Turconi and colleagues). This may be confusing, as the cardinal distance effect is generally considered a marker of magnitude processing; and an inverse distance effect would have been expected for order processing.

o In line 59, the reverse distance effect is introduced but should be clearly distinguished from the cardinal distance effect with regard to their origin.

o In line 68, the canonical distance effect is reintroduced, but in the context of order processing (for unordered sequences).

My suggestion would be to gather all the information about distance effects to clarify what it represents in each context (magnitude comparisons, ordered and unordered sequences).

We tried to clarify the distinction between the canonical and reverse distance effects in our revision with several changes. First, we now explicitly mention the canonical distance effect in our summary of the study by Turconi et al. (2004):

“However, since tasks in both conditions involved only two numbers, it is possible that participants used magnitude comparisons (as suggested by the canonical distance effect) in the ordinality condition and vice versa [19].”

Second, we added the following sentences to the section that introduces the reverse distance effect:

“A reverse distance effect also emerges for other ordinal sequences such as the letters of the alphabet or months of the year (e.g., [9,29]). This finding helps to differentiate it from the canonical distance effect associated with magnitude processing, because non-numeric sequences only convey positional information.”

Finally, we adjusted the following sentence to summarize the information at the end of this discussion:

“Overall, much of our existing knowledge about ordinal processing is linked to the three-item order verification task [4,8–10,25], suggesting that several different components are involved in this task. This is reflected by the reverse distance effect as a marker for efficient retrieval of positional information in ordered sequences, as well as the canonical distance effect as a correlate of magnitude processing in unordered sequences.”

- In line 66, authors suggest that «individuals process ordinal sequences in qualitatively distinct ways», which qualitatively distinct ways?

We have adapted this sentence to be more precise: “One possible explanation for this inconsistency might be that individuals process ordered sequences with different strategies, as not all individuals exhibit a reverse distance effect (some even show a canonical distance effect) when processing ordered sequences [25].”

- I agree that the question of whether (and how) order processing and arithmetic skills are associated is relevant. However, I think more emphasis could be placed on why order, in particular, might be a crucial element for arithmetic. For example, it could be mentioned that learning the order of the first number words has been shown to be a necessary step in the acquisition of counting and cardinal knowledge (Wynn, 1992).

We have included the following sentence in the introduction to address the relevance of order processing in the context of arithmetic: “Besides the well-known relevance of learning the correct order of number words for the development of arithmetic skills [14], current research suggests that more sophisticated skills like arithmetic rely on a rich semantic network of numerical associations, and that ordinal processing is a key indicator of this development [3].”

Methods:

- It is not mentioned why the authors excluded left-handed participants?

Left-handedness has been associated with differences in brain activations related to motor control, vision, language, and working memory (see e.g. Sha et al., 2021). There is no clear evidence if such differences would also arise in the context of our ordinality verification task, but since only about 10% of the population are left-handed, we decided to exclude left-handed people just to be on the safe side. This is arguably not ideal, but common practice to control for handedness in neuroscientific studies. Therefore, we decided not to explicitly state the rationale behind excluding left-handed individuals.

Sha Z, Pepe A, Schijven D, Carrión-Castillo A, Roe JM, Westerhausen, R, Joliot M, Fisher SE, Crivello F, Francks C. Handedness and its genetic influences are associated with structural asymmetries of the cerebral cortex in 31,864 individuals. PNAS 2021;118(47): e2113095118. doi:10.1073/pnas.2113095118

- How many channels were interpolated? How many epochs excluded? Please mention it in the manuscript.

We added this information to the beginning of the “ERP data” results section:

Preprocessing

On average, we interpolated 5.3 ± 2.6 (mean ± standard deviation) bad channels per participant (individual channel selections are available in the data repository). Furthermore, we dropped 12.2 ± 12.9 (mean ± standard deviation) epochs (from a total of 480) per participant.

- I understand why the authors decided to analyze their EEG results in a data-driven manner, but I wonder if this might have biased the results. What if the authors analyzed their results in the same regions or time segments as those defined in Rubinsten and colleagues' study?

The two earlier studies mentioned in our manuscript either used a task that is not suited to assess specific effects of ordinal processing or a different stimulus format (dot arrays). Consequently, we believe that adopting a data-driven approach is justified, especially because it includes all regions and the entire time course. Notably, the regions and time intervals identified in the studies by Turconi et al. (2004) and Rubinsten et al. (2013) did not reveal meaningful differences in our study.

- Related to that, why do the authors think that the results for non-symbolic dot arrays would be so different from those for symbolic Arabic numerals with regard to order processing? Could the authors justify why they think order processing is format-specific and not a more general mechanism? A similar question was discussed in a recent study: Wong, B., Bull, R., Ansari, D., Watson, D. M., & Liem, G. A. D. Order processing of number symbols is influenced by direction, but not format. Q. J. Exp. Psychol. 75, 98–117 (2021).

Prior behavioral and neuroimaging research indicates that order processing of dot arrays differs fundamentally from order processing of symbolic number sequences (e.g., Lyons et al., 2016, 2017; Wilkey & Ansari, 2020). The former relies on iterative magnitude comparison processes, whereas the latter is associated with the reverse distance effect (for example, see Vogel et al., 2019). The study by Wong et al. (2021) compared different symbolic formats (Arabic numerals, English number words, and Chinese number words). To the best of our knowledge, there is no EEG study that has investigated this question yet, so it might be the case that different symbolic number formats overlap significantly. However, the work by Wong et al. (2021) does not challenge our argument that dot arrays are associated with different processing mechanisms compared to number symbols.

1. 

Wilkey ED, Ansari D. Challenging the neurobiological link between number sense and symbolic numerical abilities. Annals of the New York Academy of Sciences. 2020; 1464: 76–98. doi:10.1111/nyas.14225

Results:

- Response time: What is the percentage of correct trials? Please report it.

We are not entirely sure that we understood your question, but the percentage of correct trials is the complement of the error rate, which we report in the subsequent section “Error rate”. Specifically, we already include the following statement in this section: “Overall error rates varied between 1% and 34% between individuals.” More details, including error rates grouped by condition, are available in the remainder of this section.

- Response time: Please report statistics for the no difference between counterbalanced key/answer mapping groups.

We performed both ANOVAs (one for RT and one for ERR as dependent variables) from our main analysis with the factor “group” included, and all model terms involving this factor were not significant:

Response: RT

 num Df den Df MSE F ges Pr(>F) 

group 1 71 0.208916 1.8421 0.024418 0.1790 

order 1 71 0.004517 192.0847 0.053410 < 2.2e-16 ***

group:order 1 71 0.004517 0.5408 0.000159 0.4645 

distance 1 71 0.001933 99.3152 0.012332 3.962e-15 ***

group:distance 1 71 0.001933 1.1479 0.000144 0.2876 

order:distance 1 71 0.001198 186.1277 0.014290 < 2.2e-16 ***

group:order:distance 1 71 0.001198 0.1503 0.000012 0.6994 

Response: ERR

 num Df den Df MSE F ges Pr(>F) 

group 1 71 2.49320 0.0235 0.000183 0.8785 

order 1 71 0.69051 18.6655 0.038599 4.974e-05 ***

group:order 1 71 0.69051 1.4639 0.003139 0.2303 

distance 1 71 0.58621 32.9287 0.056720 2.177e-07 ***

group:distance 1 71 0.58621 0.9400 0.001714 0.3356 

order:distance 1 71 0.75151 60.3567 0.123802 4.482e-11 ***

group:order:distance 1 71 0.75151 0.1943 0.000455 0.6607 

We have now included these results in the supplementary material, and we refer to these tables in the corresponding sentence as follows: “Furthermore, we found no differences between the two counterbalanced key/answer mapping groups, so we will ignore this factor in all subsequent analyses (see S2 and S3 Tables).”

- Unless I missed it, authors did not mention whether they corrected for multiple comparisons in their correlation analyses.

We did not correct our correlation analyses for multiple comparisons, mainly because our study is exploratory in nature with the primary goal of creating new hypotheses, which should be confirmed in follow-up studies. We clearly mention this in several locations, including a comprehensive paragraph on limitations in the conclusion section and, following a suggestion by the first reviewer, now also in the abstract.

Discussion:

- While I acknowledge the novelty of this study and recognize the potential challenges in directly comparing it to previous research, I recommend incorporating EEG literature related to the processing of Arabic digits. This inclusion would contribute to situating the present findings within the broader context of our understanding of the neurophysiological aspects associated with symbolic numeral coding.

We included the following sentence to embed our work into a broader EEG literature context: “While our findings align with previous ERP studies that have identified frontal and parietal electrodes within different time windows ranging from 120 ms to 430 ms in the context of semantic processing (e.g., [41–43]) and from 350 ms to 900 ms in the context of stimulus and response conflict (e.g., [44]), additional research is required to validate our findings and assumptions.”

- I feel that the authors simplify the discussion regarding the ERPs and correlations with arithmetic:

o Electrode clusters are very sparse (e.g., C2, AF7 and P9), and the time window selected for ordered versus unordered sequences is quite late compared to previous studies (210-240ms in Turconi et al. ; 130-200ms in Rubinsten et al. ; 500-750ms in this study). The authors should further discuss these points.

The variation in time windows can be attributed to the different tasks used in the three studies. Turconi et al. (2004) used a task that can be solved via magnitude comparison, whereas Rubinsten et al. (2013) used a sequence of three dot arrays. The time window identified in Turconi et al. (2004) is not based on an ordered versus unordered contrast (which is not possible with their stimulus set), so we cannot directly compare it with the time window of 500–750 ms in our study. We do discuss the different time windows related to distance effects (for both prior studies) in the following paragraphs though.

o How do the authors interpret the variation in mean amplitude between ordered and unordered sequences, considering that it is sometimes higher (at C2) and sometimes lower (at AF7 and P9) depending on the selected channels? Does the fact that the polarity of the order effect changes depending on the electrode suggest the involvement of different mechanisms in the processing of numerical order? If so, I believe this should be included in the discussion.

o I have the same question about the distance effects, which appear either positive (at PO8 for ordered and F8 for unordered sequences) or negative (at Fz for ordered and AFz for unordered sequences) depending on the electrodes. How do the authors interpret these changes in polarity? Are there any other ERP studies on the magnitude distance effect that reported a comparable contrasting difference at distinct sites?

Peaks in the average waveform do not hold special meaning, as they emerge from the summation of multiple underlying components. Furthermore, whether a peak is positive or negative does not provide any insights into neurophysiological processes, given that polarity also critically depends on the selected reference (in our study, the average reference). For a detailed overview, see e.g. Chapter 2 in Luck (2014).

o How do the authors interpret the significant negative correlation between arithmetic fluency and mean ERP amplitude of ordered versus unordered sequences at AF7 (where mean amplitude for ordered sequences was lower than for unordered sequences)?

Although we would like to be more specific in the interpretation of the observed correlation, we believe that our current knowledge is insufficient to make such claims for the following reasons:

1. Given that the majority of prior research has focused on the reverse distance effect, we do not know which cognitive mechanisms are engaged in the contrast of ordered versus unordered sequences.

2. The arithmetic fluency task measures a mixture of domain-general (e.g, working memory) and domain-specific (e.g., number knowledge) factors, so it is difficult to isolate the factor(s) that might relate to the observed association.

3. As discussed in the previous response, it is not straightforward to interpret the neurophysiological sources underlying the significant negative correlation.

4. Our finding only suggests a mild association between mean ERP amplitude and arithmetic fluency, so we would prefer to replicate this pattern in a follow-up study before making claims about the potential mechanisms.

---

## [Decision Letter · Decision Letter 1]

12 Mar 2024

Electrophysiological correlates of symbolic numerical order processing

PONE-D-23-37780R1

Dear Dr. Brunner,

We’re pleased to inform you that your manuscript has been judged scientifically suitable for publication and will be formally accepted for publication once it meets all outstanding technical requirements.

Kind regards,

Jérôme Prado

Academic Editor

PLOS ONE

Additional Editor Comments (optional):

Reviewers' comments:

Reviewer's Responses to Questions

**Comments to the Author**

1. If the authors have adequately addressed your comments raised in a previous round of review and you feel that this manuscript is now acceptable for publication, you may indicate that here to bypass the “Comments to the Author” section, enter your conflict of interest statement in the “Confidential to Editor” section, and submit your "Accept" recommendation.

Reviewer #1: All comments have been addressed

Reviewer #2: All comments have been addressed

2. Is the manuscript technically sound, and do the data support the conclusions?

Reviewer #1: Yes

Reviewer #2: Yes

3. Has the statistical analysis been performed appropriately and rigorously? 

Reviewer #1: Yes

Reviewer #2: Yes

4. Have the authors made all data underlying the findings in their manuscript fully available?

Reviewer #1: Yes

Reviewer #2: Yes

5. Is the manuscript presented in an intelligible fashion and written in standard English?

Reviewer #1: Yes

Reviewer #2: Yes

6. Review Comments to the Author

Reviewer #1: All my previous comments have been addressed by the authors in this revision. I have no further comments.

Reviewer #2: The authors have thoroughly addressed all concerns raised by the reviewers. With these additional explanations, the manuscript offers sufficient detail to comprehend the research, its hypotheses, and the conclusions presented herein.

7. PLOS authors have the option to publish the peer review history of their article (what does this mean?). If published, this will include your full peer review and any attached files.

Reviewer #1: **Yes: **Declan Devlin

Reviewer #2: No

---

## [Editor Report · Acceptance letter]

14 Mar 2024

PONE-D-23-37780R1 

PLOS ONE

Dear Dr. Brunner, 

I'm pleased to inform you that your manuscript has been deemed suitable for publication in PLOS ONE. Congratulations! Your manuscript is now being handed over to our production team.

Kind regards, 

on behalf of

Dr. Jérôme Prado 

Academic Editor

PLOS ONE